# Mixture of Experts Characteristic Function Embeddings for Heterogeneous Fraud Graphs

## Abstract

Fraud detection over heterogeneous graphs requires reasoning over multiplex relations, attribute polymorphism, and structural heterophily, yet prevailing detectors entangle these orthogonal biases into monolithic pipelines assumed to generalize across domains. We address this limitation by instantiating a decouple–then–fuse representation learning that isolates structural and attribute channels before reintroducing interaction through an adaptive fusion interface. On the structural side, we encode distributional neighborhood context via characteristic-function signatures compressed through randomized spectral factorization; on the attribute side, we deploy input-adaptive Mixture-of-Experts projections that specialize each instance to role-conditioned patterns. The two views are subsequently reconciled through a Bayesian mean–difference fusion layer that models per-node consensus and discrepancy, enabling calibrated integration under heterophily and cross-modal conflict. Empirical evaluation across benchmark fraud graphs from telecom records, e-commerce reviews, and cryptocurrency transactions domains shows improved fraud node detection performance, attributable to the model's ability to disentangle structural and attribute features and reconcile their discrepancies.

## 1 Introduction

Across diverse operational domains including telecommunications call-detail networks Becker et al. (2010); Edozie et al. (2025); Pourhabibi et al. (2020), heterogeneous e-commerce review interaction graphs Carta et al. (2019), and blockchain transaction ledgers Weber et al. (2019); Asiri & Somasundaram (2025) fraud arises from relational structures intertwined with high-dimensional attribute spaces that span multiple behavioral patterns. These domains instantiate heterogeneous fraud graphs, wherein heterogeneity encompasses (i) relational multiplexity, i.e., multiple edge types, asymmetric link semantics Breit et al. (2023); (ii) attribute polymorphism, i.e., multi-modal feature distributions corresponding to distinct role classes such as illicit actors, benign participants, and intermediary conduits Pandit et al. (2007); and (iii) structural heterophily, wherein anomalous nodes preferentially attach to dissimilar others, violating the homophily priors that undergird many representation learning models Platonov et al. (2023). Each of these axes imposes orthogonal inductive biases: relational multiplexity demands relation-specific propagation operators or meta-path formulations Schlichtkrull et al. (2018), attribute polymorphism benefits from feature-aware nonlinear projections Jacobs et al. (1991a), and structural heterophily necessitates distributional structural encodings that are invariant to neighbor label discordance Ribeiro et al. (2017).

The literature on fraud detection over such graphs has consequently fragmented into modality-specialized approaches. On the structural side, meta-path guided relation encoders Vrahatis et al. (2024) and spectral or subgraph motif embeddings Hooi et al. (2016); Monti et al. (2018) have sought to encode multiplex relational patterns. Attribute-centric models, conversely, have focused on supervised or semi-supervised projections Lebichot et al. (2016); Xiang et al. (2023), cost-sensitive classifiers Hu et al. (2024) to amplify the feature signal carried by rare positive labels in high-dimensional feature spaces. Heterophily-robust graph methods have emerged that modulate message passing via predicted label compatibility matrices Zhu et al. (2021) or filter neighbor contributions through learned structural gates Innan et al. (2024). While effective in their respective niches, these methods tend to entangles modality-specific inductive biases within monolithic architectures, which risk overfitting to the statistical characteristics of a given domain (e.g., telecom vs. blockchain transaction topology) and hinders cross-domain transferability Zhu et al. (2020b); Chen et al. (2024).

The absence of a unifying representation learning scheme capable of decoupling and subsequently reconciling structural and attribute signals under arbitrary combinations of heterogeneity remains a bottleneck Mondal et al. (2025). A generalized framework would ideally (a) instantiate modality-specific encoders with sufficient capacity to specialize to the heterogeneity axis they address; (b) maintain a clean separation between structural and attribute processing paths to avoid mutual contamination of inductive biases; and (c) fuse the resulting codes via a mechanism that can explicitly model agreement and disagreement, thereby exposing the interaction surface between modalities as an additional discriminative feature space. Such a construction affords compositionality Koutra et al. (2015): advances in, e.g., heterophily-resilient structural summarization or pattern-aware attribute encoding can be incorporated without re-architecting the entire pipeline.

Several precursors to this model target exist. Mixture-of-Experts (MoE) architectures Jacobs et al. (1991b) have demonstrated the utility of sparse, input-adaptive expert routing in many domains, suggesting analogous benefits for attribute polymorphism in fraud graphs Shazeer et al. (2017). Characteristic-function embeddings of node neighborhoods Rozemberczki & Sarkar (2020) provide a permutation-invariant, distributional view of structural context, naturally robust to neighbor-label discordance, and are amenable to low-rank compression via truncated SVD Qiu et al. (2018); Chanpuriya et al. (2020) for scalability on million-node graphs. Bayesian-inspired fusion operators Kittler (1998); Bach & Jordan (2005); Atrey et al. (2010) that combine modality-specific encodings under uncertainty weighting have shown promise in multi-sensor and multi-view learning, but have yet to be systematically instantiated for fraud graphs with explicit per-node adaptivity.

We posit that the confluence of these three components (i) per-node adaptive attribute specialization via MoE, (ii) structure-aware neighborhood signatures via characteristic functions and spectral compression, and (iii) conflict-aware Bayesian mean–difference fusion constitutes a domain-agnostic, heterogeneity-resilient representation learning. The work makes the contributions:

- **Decouple–then–fuse representation learning.** We propose a decouple–then–fuse representation learning that explicitly separates how the model processes node attributes and graph structure, avoiding premature entanglement of their differing inductive assumptions, and fusing them later through a calibrated fashion. This disentanglement addresses a limitation of prior approaches that entangle structural and attribute features, leading the model to learn mixed, non-discriminative representations when fraudulent nodes deviate from neighborhood patterns leading to degraded performance under structural heterophily, i.e., a method may perform well on one fraud graph but fail to adapt to another due to differences in structural patterns and relational biases.

- **Conflict-solving adaptive fusion.** We employ a calibrated fusion method that reconciles structural and attribute representations by explicitly modeling their consensus and discrepancy on a per-node basis. This adaptive integration solves conflicts that arise when structure and attributes provide contradictory patterns through a per-node consensus difference fusion that first forms the discrepancy of the two views (structural and attributive), and then applies a learned per-node weight, predicted by a variational mixer, to scale the discrepancy before composition. The resulting representations generalize across domains, as validated on four real-world fraud graphs spanning telecom records, e-commerce reviews, and cryptocurrency transactions.

## 2 RELATED WORK

The literature on fraud detection over relational data has evolved along parallel trajectories, i.e., domain-specific studies that engineer bespoke heuristics and learning pipelines for their operational context, and more recent efforts that seek to abstract these advances into domain-agnostic graph representation frameworks capable of accommodating label sparsity and structural heterophily. Representative works include feature disguise graph fraud detectors and samplers Liu et al. (2021); Huang et al. (2022), heterogeneous/relational encoders that operationalize meta-paths and dense subgraph patterns Hooi et al. (2016); Dong et al. (2017), heterophily-resilient propagation via ego/neighbor decoupling or frequency-adaptive filters Zhu et al. (2020a); Bo et al. (2021); Zhu et al. (2021).

Label-efficiency and generalization are pursued via self-supervised/contrastive pretraining on graphs Veličković et al. (2018); Hu et al. (2020) and positive-unlabeled or cost-sensitive risk formulations for rare positives Sahin et al. (2013); Ma et al. (2021); Khodabandehlou & Golpayegani (2024). Further, cost-sensitive ensembles that couple graph attention networks with boosting mitigate se-

vere class imbalance by propagating misclassification costs through the embedding updates Hu et al. (2024). Fault detection has been cast as bi-level graph reasoning that hierarchically encodes inter-node topology alongside intra-node software-execution traces, while semi-supervised methods modulate message passing via layer-wise label predictions to differentially gate aggregation under heterophily Bourgerie & Zanouda (2023); Hyun et al. (2024).

Graph detectors inherit message-passing biases, i.e., low-pass smoothing and homophily-leaning feature mixing, deteriorate under heterophily, structural discordance, or adversarial rewiring Zhu et al. (2020a); Rusch et al. (2023). Decoupling propagation from representation incorporate: encode topology via permutation-invariant, higher-order distributional summaries, e.g., characteristic-function neighborhood signatures, that are compact and noise-robust Qiu et al. (2018); Rozember-czki & Sarkar (2020), and reserving the attribute channel for sparse, input-adaptive MoE projections that specialize by role/pattern to improve the discriminative performance Wang et al. (2023); Yang et al. (2024). To our knowledge, no prior work jointly couples distributional structural encodings with MoE-based attribute specialization under an explicit fusion.

# 3 MIXTURE OF EXPERTS AUGMENTED CHARACTERISTIC FUNCTION GRAPH EMBEDDING

## 3.1 NOTATION AND PROBLEM SETTING

Let $G = (V, E)$ be an undirected, simple graph with $N = |V|$ nodes and adjacency matrix $\mathbf{A} \in \{0, 1\}^{N \times N}$. Each vertex $v_i \in V$ is endowed with a raw attribute vector $\mathbf{x}_i \in \mathbb{R}^d$ and the entire attribute matrix is written $\mathbf{X} = [\mathbf{x}_1^\top, \ldots, \mathbf{x}_N^\top]^\top \in \mathbb{R}^{N \times d}$. In the context, $N$ denotes the number of nodes, $d$ the number of attributes per node, $\mathbf{x}_i \in \mathbb{R}^d$ the attribute vector of node $v_i$. Each node carries a ground-truth label $y_i \in \mathcal{Y}$ (typically binary in fraud datasets), and we denote by $\mathcal{L} \subseteq V$ the subset of vertices whose labels are observed during training.

## 3.2 LOCAL ATTRIBUTE ENCODING VIA A MIXTURE OF EXPERTS

The initial objective is to transform $\mathbf{x}_i$ into a task-aligned representation $\mathbf{z}_i \in \mathbb{R}^{d'}$ that can later be fused with structural information. Inspired by the routing–by-agreement paradigm in deep mixture-of-experts (MoE) models Jacobs et al. (1991b); Shazeer et al. (2017), we instantiate $M$ domain-specialised experts, each realised as a two-layer perceptron:

$$\mathbf{e}_i^{(m)} = \phi\big(\mathbf{W}_2^{(m)} \sigma(\mathbf{W}_1^{(m)}\mathbf{x}_i + \mathbf{b}_1^{(m)}) + \mathbf{b}_2^{(m)}\big) \in \mathbb{R}^{d'}, \tag{1}$$

where $\sigma(\cdot)$ and $\phi(\cdot)$ denote ReLU and identity activations, respectively, and $\{\mathbf{W}_1^{(m)}, \mathbf{b}_1^{(m)}, \mathbf{W}_2^{(m)}, \mathbf{b}_2^{(m)}\}$ collect the trainable parameters of the $m$-th expert.

A routing network $\mathbf{W}_r \in \mathbb{R}^{M \times d}$ assigns probabilistic weights

$$w_{i,m} = \text{softmax}_m\big(\mathbf{W}_r\mathbf{x}_i + \mathbf{b}_r\big), \qquad \sum_{m=1}^{M} w_{i,m} = 1, \tag{2}$$

which quantify the affinity between node $v_i$ and expert $m$. To avoid excessive computational overhead, we retain only the $k$ largest routing weights, collected in the index set $\mathcal{K}_i = \arg\text{top}_k\{w_{i,m}\}_{m=1}^M$. The MoE feature for $v_i$ is the convex combination

$$\mathbf{z}_i = \sum_{m \in \mathcal{K}_i} \frac{w_{i,m}}{\sum_{m' \in \mathcal{K}_i} w_{i,m'}} \mathbf{e}_i^{(m)}. \tag{3}$$

Collecting over $i$ yields the matrix $\mathbf{Z} = [\mathbf{z}_1^\top, \ldots, \mathbf{z}_N^\top]^\top \in \mathbb{R}^{N \times d'}$.

**Training objective.** We attach a task head $f : \mathbb{R}^{d'} \to \mathbb{R}^{|\mathcal{Y}|}$ (a single linear layer) to every $\mathbf{z}_i$ and optimise all MoE parameters $\{\mathbf{W}_1^{(m)}, \mathbf{b}_1^{(m)}, \mathbf{W}_2^{(m)}, \mathbf{b}_2^{(m)}, \mathbf{W}_r, \mathbf{b}_r\}$ together with $f$ by minimising

$$\mathcal{L} = \sum_{i \in \mathcal{L}} \ell\big(f(\mathbf{z}_i), y_i\big),$$

where $\ell$ denotes the cross-entropy term. After convergence, $f$ is discarded and the frozen encoder provides the attribute embedding $\mathbf{Z}$ used in subsequent fusion.

The gating mechanism in equation 3 equips each node with an input-dependent set of specialist projections, allowing the encoder to target its representation capacity to heterogeneous node populations (e.g. fraudulent, benign, and courier entities in our task), while the top-$k$ sparsification keeps both time and memory requirements bounded, detailed with targeted experiments in Appendix A.2.

### 3.3 NEIGHBOURHOOD SIGNATURE BY CHARACTERISTIC FUNCTIONS

To capture higher-order relational signals we compute, for every node, a vector that summarises the distribution of attributes inside its random-walk neighbourhood. Let $\mathbf{D} = \mathrm{diag}(\deg v_1, \ldots, \deg v_N)$ and define the row-normalised walk matrix $\tilde{\mathbf{A}} = \mathbf{D}^{-1}\mathbf{A}$. Given a set of evaluation points $\Theta = \{\theta_\ell\}_{\ell=1}^{L} \subset (0, \theta_{\max}]$, we first lift raw attributes into a Fourier-like feature space:

$$\mathbf{\Phi} = \big[\cos(\mathbf{X} \otimes \Theta) \mid \sin(\mathbf{X} \otimes \Theta)\big] \in \mathbb{R}^{N \times 2dL}, \tag{4}$$

where "$\otimes$" denotes the outer product applied row-wise. Starting with $\mathbf{H}^{(0)} = \mathbf{\Phi}$ we perform $O$ diffusion steps $\mathbf{H}^{(t)} = \tilde{\mathbf{A}}\mathbf{H}^{(t-1)}$ and store each intermediate block. The concatenation $\mathbf{H} = [\mathbf{H}^{(1)}\|\ldots\|\mathbf{H}^{(O)}] \in \mathbb{R}^{N \times 2dLO}$ encodes successively expanding random-walk neighbourhoods.

**Spectral compression via truncated SVD.** Let $\mathbf{H} \in \mathbb{R}^{N \times 2dLO}$ be the walk–feature matrix, where $d$ is the number of raw node attributes, $L$ the number of characteristic-function evaluation points, and $O$ the maximum walk order. Following Rozemberczki & Sarkar (2020), we set $L, O$ (25, 5) and keep only the leading $r$ singular directions ($r$ is the target embedding dimension) by approximating

$$\mathbf{H} \approx \hat{\mathbf{H}} = \mathbf{U}_r \mathbf{S}_r \mathbf{V}_r^{\top},$$

$$\mathbf{U}_r \in \mathbb{R}^{N \times r}, \quad \mathbf{S}_r = \mathrm{diag}(s_1, \ldots, s_r), \quad \mathbf{V}_r \in \mathbb{R}^{(2dLO) \times r}.$$

**Computation.** We use a randomized Lanczos routine: an efficient variant of power iteration that builds a low-dimensional Krylov subspace Liesen & Strakos (2013) to capture the dominant spectrum. The hyper-parameter $\eta$ (we use $\eta = 2$) is the number of Krylov iterations, i.e. the number of times the algorithm repeatedly multiplies a random sketch by $\tilde{\mathbf{A}}$ to refine the subspace. Because $\mathbf{H}$ is accessed only through such multiplications, the overall cost is $O\big((N+E)\,r\,\eta\big)$ in time and $O(Nr)$ in memory. Node-level code is given by the retained factor

$$\mathbf{E} = \mathbf{U}_r \mathbf{S}_r \in \mathbb{R}^{N \times r}$$

minimises $\|\mathbf{H} - \hat{\mathbf{H}}\|_F$ over all rank-$r$ reconstructions, preserving a fraction $\big(\sum_{j=1}^{r} s_j^2\big)/\big(\sum_j s_j^2\big)$ of the total signal energy. Orthogonality of the columns of $\mathbf{U}_r$ yields uncorrelated axes, and the low rank acts as an information bottleneck that curbs noise and overfitting properties, mirroring the behaviour reported in Rozemberczki & Sarkar (2020).

### 3.4 BAYESIAN MEAN–DIFFERENCE ATTRIBUTE FUSION WITH ADAPTIVE $\lambda$

We fuse the raw attributes $\mathbf{E}$ and MoE embeddings $\mathbf{Z}$ into a single attribute representation via a Consensus–Residual Fusion (CRF) with per-node mixing inspired by Bach & Jordan (2005); Papananias et al. (2022). The two views are aligned by projecting each onto its top $d^\star = \min(d, d')$ principal components, followed by column-wise whitening to unit variance. Let $\mathbf{P}_E \in \mathbb{R}^{d \times d^\star}$ and $\mathbf{P}_Z \in \mathbb{R}^{d' \times d^\star}$ denote the PCA projection matrices fitted on $\mathbf{E}$ and $\mathbf{Z}$, respectively. Define aligned, whitened features:

$$\tilde{\mathbf{E}} = \mathrm{whiten}(\mathbf{E}\mathbf{P}_E), \qquad \tilde{\mathbf{Z}} = \mathrm{whiten}(\mathbf{Z}\mathbf{P}_Z),$$

where $\mathrm{whiten}(\cdot)$ rescales each column to unit empirical standard deviation.

**Bayesian averaging and discrepancy.** Estimate a shared precision $\tau = \sigma^{-2}$ from the empirical discrepancy $\sigma^2 = \frac{1}{Nd^\star} \sum_{i=1}^{N} \|\tilde{\mathbf{e}}_i - \tilde{\mathbf{z}}_i\|_2^2$. For node $i$,

$$\mathbf{h}_i = \frac{\tau \tilde{\mathbf{e}}_i + \tau \tilde{\mathbf{z}}_i}{2\tau} = \tfrac{1}{2}(\tilde{\mathbf{e}}_i + \tilde{\mathbf{z}}_i), \qquad \mathbf{d}_i = \tilde{\mathbf{z}}_i - \tilde{\mathbf{e}}_i \in \mathbb{R}^{d^\star}.$$

Given a mixing coefficient $\lambda \geq 0$, the fused attribute vector is the mean–difference concatenation

$$\mathbf{f}_i^{(\lambda)} = \big[\, \mathbf{h}_i^{\top} \mid (\lambda\, \mathbf{d}_i)^{\top} \,\big]^{\top} \in \mathbb{R}^{2d^\star}. \tag{5}$$

## 3.5 PER-NODE VARIATIONAL MIXING AND ADAPTIVE ATTRIBUTE FUSION

Existent fraud graphs exhibit heterogeneous agreement between the structural view $\tilde{\mathbf{e}}_i \in \mathbb{R}^{d^\star}$ (aligned and whitened from $\mathbf{E}$) and the MoE view $\tilde{\mathbf{z}}_i \in \mathbb{R}^{d^\star}$ (aligned and whitened from $\mathbf{Z}$); this heterogeneity makes a single global mixing weight suboptimal. We therefore learn, for each node $v_i$, a weight coefficient that adjusts how much of the MoE-specific discrepancy to inject into the final representation.

**Latent consensus–difference mixer.** To separate shared signal from view-specific residue, we introduce a consensus latent $\mathbf{c}_i \in \mathbb{R}^p$ and a discrepancy latent $\mathbf{d}_i \in \mathbb{R}^p$ ($p{=}16$), and parameterise amortised Gaussian posteriors:

$$q_e(\mathbf{c}_i \mid \tilde{\mathbf{e}}_i) = \mathcal{N}(\boldsymbol{\mu}_e, \operatorname{diag} \boldsymbol{\sigma}_e^2), \qquad q_z(\mathbf{u}_i \mid \tilde{\mathbf{z}}_i) = \mathcal{N}(\boldsymbol{\mu}_z, \operatorname{diag} \boldsymbol{\sigma}_z^2), \quad \mathbf{u}_i = \mathbf{c}_i + \mathbf{d}_i.$$

Via the reparameterisation trick, we obtain samples

$$\mathbf{c}_i = \boldsymbol{\mu}_e + \boldsymbol{\sigma}_e \odot \boldsymbol{\epsilon}_e, \qquad \mathbf{d}_i = (\boldsymbol{\mu}_z + \boldsymbol{\sigma}_z \odot \boldsymbol{\epsilon}_z) - \mathbf{c}_i,$$

so that $\mathbf{c}_i$ captures view agreement and $\mathbf{d}_i$ isolates MoE-only corrections. A linear gate then outputs a per-node trust weight:

$$\lambda_i = \sigma(\mathbf{w}^\top \mathbf{d}_i) \in (0,1), \qquad \mathbf{f}_i = \left[\, \mathbf{c}_i^\top \mid (\lambda_i \mathbf{d}_i)^\top \,\right]^\top \in \mathbb{R}^{2p},$$

where $\mathbf{f}_i$ is the task-facing fused latent. The coefficient $\lambda_i$ modulates the contribution of the discrepancy latent and thus quantifies, for node $v_i$, how much of the MoE-specific signal is deemed informative.

**Objective and learning.** The mixer is trained to be both predictive, yielding label-aligned discriminability, and well-posed, i.e., constrained by reconstruction consistency and variational regularisation for identifiable latents and stable posteriors, which yields reproducible $\lambda_i$ estimates with calibrated uncertainty, and supports reliable generalisation. Predictivity is enforced by a supervised head $h$ acting on $\mathbf{f}_i$ with cross-entropy loss $\operatorname{CE}(y_i \mid \mathbf{f}_i)$. Well-posedness is enforced by an ELBO-inspired objective that tethers latents to their encoders and regularises posteriors:

$$\mathcal{L} = \underbrace{\operatorname{CE}(y_i \mid \mathbf{f}_i)}_{\text{task}} + \underbrace{\|\mathbf{c}_i - \boldsymbol{\mu}_e\|_2^2 + \|(\mathbf{c}_i + \mathbf{d}_i) - \boldsymbol{\mu}_z\|_2^2}_{\text{consistency (Gaussian likelihood after whitening)}} + \beta\left[\operatorname{KL}\!\big(q_e\|\mathcal{N}(\mathbf{0},\mathbf{I})\big) + \operatorname{KL}\!\big(q_z\|\mathcal{N}(\mathbf{0},\mathbf{I})\big)\right],$$

The consistency terms coincide with negative log-likelihoods under isotropic Gaussians induced by whitening, while the KL terms prevent degenerate posteriors. Joint optimisation over encoder parameters and $\mathbf{w}$ shapes $\lambda_i$ to upweight discrepancies that improve classification and downweight those that behave like noise.

**Deterministic inference and closed-form composition.** At test time, we use posterior means for stability,

$$\hat{\mathbf{c}}_i = \boldsymbol{\mu}_e(\tilde{\mathbf{e}}_i), \qquad \hat{\mathbf{d}}_i = \boldsymbol{\mu}_z(\tilde{\mathbf{z}}_i) - \hat{\mathbf{c}}_i, \qquad \hat{\lambda}_i = \sigma(\mathbf{w}^\top \hat{\mathbf{d}}_i).$$

The resulting trust weight is then applied to a closed-form mean–difference fuse in observation space, which retains the interpretability of the characteristic-function view: with

$$\mathbf{h}_i = \tfrac{1}{2}\big(\tilde{\mathbf{e}}_i + \tilde{\mathbf{z}}_i\big), \qquad \mathbf{d}_i^{(\mathrm{obs})} = \tilde{\mathbf{z}}_i - \tilde{\mathbf{e}}_i,$$

we define the per-node fused attribute vector:

$$\mathbf{F}_i^{(\hat{\lambda}_i)} = \left[\, \mathbf{h}_i^\top \mid (\hat{\lambda}_i\, \mathbf{d}_i^{(\mathrm{obs})})^\top \,\right]^\top \in \mathbb{R}^{2d^\star}.$$

This composition preserves the semantics of agreement ($\mathbf{h}_i$) and disagreement ($\mathbf{d}_i^{(\mathrm{obs})}$) while letting the variational mixer decide how strongly the disagreement should enter for each node.

**Final representation and structural fusion.** The per-node fused vectors are stacked as $\mathbf{F}^{(\hat{\lambda})} = \big[\mathbf{f}_1^{(\hat{\lambda}_1)\top}, \ldots, \mathbf{f}_N^{(\hat{\lambda}_N)\top}\big]^\top \in \mathbb{R}^{N \times 2d^\star}$ and concatenated with the raw structural embedding $\mathbf{E} \in \mathbb{R}^{N \times r}$ (Section 3.3):

$$\mathbf{F} = \left[\, \mathbf{E} \mid \mathbf{F}^{(\hat{\lambda})} \,\right] \in \mathbb{R}^{N \times (r + 2d^\star)}.$$

This construction keeps explicit structural coordinates $\mathbf{E}$ alongside the consensus–discrepancy statistics captured by $\mathbf{F}^{(\hat{\lambda})}$; the two live in complementary subspaces (global, low-rank structure vs. per-node agreement/difference features), enabling the downstream classifier to weight topology and cross-view evidence separately rather than collapsing them into a single channel.

Table 1: Statistics of the four benchmark fraud graphs. Here, $N$ denotes the number of nodes, $E$ the number of edges, and $d$ the dimensionality of each node's feature vector. The collection covers two telecom call-network datasets (SC-TF and BUPT-TF), one e-commerce review graph (YelpChi), and one cryptocurrency transaction graph (Bitcoin).

| Dataset | Domain | $N$ | $E$ | $d$ | Positive |
|---------|--------|-----|-----|-----|----------|
| SC-TF | Telecom | 6,106 | 838,528 | 55 | Fraud |
| BUPT-TF | Telecom | 116,383 | 350,751 | 39 | Fraud |
| YelpChi | E-commerce | 45,954 | 8,051,348 | 32 | Spam |
| Bitcoin | Cryptocurrency | 203,769 | 234,355 | 166 | Illicit |

**Implementation note (discretisation and caching).** For large $N$, we discretise $\hat{\lambda}_i$ onto a fine grid $\Lambda \in [0, 1]$ and cache $\{\mathbf{F}^{(\lambda)}\}_{\lambda \in \Lambda}$; at inference we index the cache with the predicted $\hat{\lambda}_i$. This yields the same limit as the continuous formulation as the bin width shrinks, while reducing recomputation. As ablations, we report (i) a fixed-$\lambda$ baseline and (ii) a discretised grid search result to isolate the contribution of adaptive mixing in Section 4.3.

## 4 EXPERIMENTS

### 4.1 DATASETS

To demonstrate the cross-domain performance of our MoE Characteristic–Function model, we conduct experiments on four real–world fraud graphs drawn from telecom, e-commerce, and cryptocurrency domains. Table 1 lists their principal statistics and brief descriptions.

**CMSC Telecom Fraud (SC-TF)** Hu et al. (2022). Released by China Mobile Sichuan Company (CMSC) during the 2019–2020 Telecom Fraud competition, SC-TF is a call–network graph of $N = 6{,}106$ mobile subscribers linked by $E = 838{,}528$ edges extracted from Call Detail Records (CDRs). Each vertex carries a $d = 55$–dimensional behavioural feature vector capturing call or SMS patterns. Labels are binary (fraudulent vs. benign), with 1 962 fraudsters (32.1%) and 4,144 normal users, yielding a markedly imbalanced setting.

**BUPT Telecom Fraud (BUPT-TF)** Liu et al. (2019). Collected by Beijing University of Posts and Telecommunications, BUPT-TF spans one week of metropolitan call activities. The graph contains $N = 116{,}383$ subscriber nodes and $E = 350{,}751$ communication edges. Raw CDR logs are distilled into $d = 39$ numerical features per node. The original annotation distinguishes fraudsters 8,448, couriers 8,074, and normal users 99,861.

**YelpChi Spam Review Graph (YelpChi)** Mukherjee et al. (2013). YelpChi is an e-commerce, social graph of $N = 45{,}954$ restaurant and hotel reviews, internally connected by around 8 million relational edges of three types: reviewer-shared (R–U–R), product-rating-shared (R–S–R), and time-cohort (R–T–R). Each review node is described by $d = 32$ handcrafted metadata and linguistic features. Binary labels mark 6,677 reviews as spam (14.5%) and 39,277 as legitimate, making YelpChi a standard benchmark for heterogeneous relation e-commerce fraud detection.

**Elliptic Bitcoin Transaction Graph (Bitcoin)** Weber et al. (2019). The Elliptic Bitcoin data set models the flow of Bitcoins as a directed transaction graph with $N = 203{,}769$ transaction nodes and $E = 234{,}355$ edges. Every node is annotated with $d = 166$ transactional features spanning local and one-hop aggregated statistics. Class labels are ternary: illicit 4,545, licit 42,019, and unknown, but we focus on the licit vs. illicit discrimination task prevalent in anti-money-laundering.

### 4.2 BASELINES

**FEATHER–N** We compare against FEATHER–N from Rozemberczki & Sarkar (2020) (as a characteristic function baseline), an attributed–neighbourhood encoder that has demonstrated competitive performance on attributed node classification for graphs. The method summarizes each node's

neighbourhood via characteristic functions of its attributes diffused by random walks and compressed by truncated SVD, making it the canonical characteristic–function baseline for our setting. We use the implementation with same configuration reported, fit it on the graphs of the aforementioned datasets, and feed the resulting embeddings to the same downstream classifier as our method for a controlled comparison.

**NetMF.**   We compare against NetMF Qiu et al. (2018), a matrix factorization approach that unifies skip-gram random-walk methods by explicitly constructing a closed-form pointwise mutual information (PMI) matrix that aggregates multi-hop co-occurrences derived from successive powers of the normalized adjacency, and then applying truncated SVD to obtain node embeddings. This formulation captures higher-order proximity without stochastic walk sampling and has shown strong performance on canonical network mining tasks. We follow the configuration reported in the paper and, for parity, feed the resulting embeddings into the same downstream classifier as our method.

**GraRep.**   We include GraRep Cao et al. (2015), a global-structure embedding method that constructs pointwise mutual information (PMI) matrices for multiple walk lengths by leveraging successive powers of the normalized adjacency, factorizes each order via truncated SVD, and concatenates the resulting per-order embeddings to capture multi-hop proximities. This matrix factorization view integrates higher-order relational context without stochastic walk sampling and has demonstrated strong performance on clustering and classification tasks. We follow the configuration reported in the original paper and feed the resulting embeddings into the same downstream classifier as our method for a controlled comparison.

**GCNSVD.**   We evaluate GCNSVD Entezari et al. (2020), a robustness-oriented 2-layer GCN that preconditions the graph via a truncated SVD, reconstructing a low-rank approximation of the adjacency before propagation to suppress high-rank (high-frequency) perturbations characteristic of adversarial rewiring. This low-rank sanitization filters spurious edges while retaining dominant structural modes, after which standard GCN message passing and classification are performed. We follow the configuration reported in the original paper and assess performance under the same data splits and metrics as other baselines. Notably, similar GCN-style architectures have been deployed on the Elliptic Bitcoin fraud dataset Asiri & Somasundaram (2025), underscoring the relevance of this baseline to our evaluation setting.

### 4.3 Results

We adopt a standardized experimental setup for controlled comparability across methods. For each dataset, the adjacency matrix is extracted from the underlying graph and node attributes are z–scored, with missing values filtered. Node embedding baselines (i.e., NetMF, GraRep, FEATHER–N) are coupled with a uniform downstream classifier implemented as a stacked feed-forward network with two hidden layers, ReLU activations, and a sigmoid output for binary discrimination. The classifier is optimized with Adam under weighted binary cross-entropy, trained with early stopping on weighted F1 to mitigate overfitting. For GCNSVD, the intrinsic graph convolutional classifier is trained under the same stopping criterion with fixed hidden dimension, dropout, and learning rate default as specified in DeepRobust (2020). All methods are executed over five independent runs with distinct random seeds, and we report mean and standard deviation for the evaluation metrics. This protocol isolates representational quality as the primary variable of interest by having the same preprocessing, splitting, and classifier architecture across the evaluation pipeline.

Across all the datasets shown in Table 2, the proposed MoECF–AF (Adaptive Fuse) establishes the relatively strongest performance, i.e., it improves the primary discrimination metric F1 while simultaneously shifting AUROC/AUPRC upward, thereby enhancing both separability and ranking fidelity (detailed in Appendix A.1). This pattern is robust to split stochasticity standard deviations are uniformly small, suggesting that the gains stem from inductive bias rather than optimization noise. In effect, decoupling structure and attributes, followed by the adaptive fusion, yields a representation that is heterophily-resilient and calibration-preserving: high recall does not come at the expense of precision in the head of the ranking. Across domains, consistent patterns emerge: on telecom graphs where neighborhood distributions are relatively stable, characteristic-function encoders already constitute strong baselines, yet MoECF–AF delivers additional uplift by contributing role-conditioned attribute specialization that complements distributional structure; on e-commerce

Table 2: Results of Node classification on four fraud graphs spanning telecom (SC–TF, BUPT–TF), e-commerce (YelpChi), and cryptocurrency (Bitcoin). Methods: GCNSVD (GCN with truncated SVD), NetMF, GraRep, FEATHER–N (characteristic–function encoder), MoECF (ours: concatenation of CF and MoE attributes), and MoECF–AF (ours: adaptive fusion of CF and MoE features). All methods use identical data splits and the same downstream classifier protocol.

| Dataset / Method | | Precision | Recall | F1 | Accuracy | AUROC | AUPRC |
|---|---|---|---|---|---|---|---|
| SC–TF | GCNSVD | **0.902** ± **0.005** | 0.837 ± 0.010 | 0.860 ± 0.006 | **0.887** ± **0.004** | 0.904 ± 0.005 | 0.886 ± 0.005 |
| | NetMF | 0.805 ± 0.006 | 0.804 ± 0.006 | 0.803 ± 0.006 | 0.804 ± 0.006 | 0.884 ± 0.006 | 0.872 ± 0.006 |
| | GraRep | 0.814 ± 0.011 | 0.811 ± 0.006 | 0.810 ± 0.006 | 0.811 ± 0.006 | 0.898 ± 0.006 | 0.909 ± 0.006 |
| | FEATHER–N | 0.869 ± 0.005 | 0.857 ± 0.001 | 0.856 ± 0.002 | 0.857 ± 0.001 | 0.928 ± 0.002 | 0.942 ± 0.002 |
| | MoECF | 0.872 ± 0.006 | 0.864 ± 0.004 | 0.863 ± 0.003 | 0.864 ± 0.004 | 0.932 ± 0.004 | 0.939 ± 0.004 |
| | MoECF–AF | 0.880 ± 0.001 | **0.873** ± **0.003** | **0.873** ± **0.003** | 0.873 ± 0.003 | **0.935** ± **0.003** | **0.944** ± **0.003** |
| BUPT–TF | GCNSVD | 0.881 ± 0.008 | 0.893 ± 0.025 | 0.887 ± 0.008 | 0.892 ± 0.025 | 0.956 ± 0.015 | 0.920 ± 0.015 |
| | NetMF | 0.825 ± 0.001 | 0.963 ± 0.001 | 0.889 ± 0.001 | 0.879 ± 0.001 | 0.888 ± 0.002 | 0.772 ± 0.002 |
| | GraRep | 0.882 ± 0.004 | **0.994** ± **0.003** | 0.935 ± 0.001 | 0.931 ± 0.001 | 0.920 ± 0.002 | 0.799 ± 0.002 |
| | FEATHER–N | 0.918 ± 0.032 | 0.965 ± 0.021 | 0.943 ± 0.011 | 0.930 ± 0.021 | 0.980 ± 0.015 | 0.978 ± 0.015 |
| | MoECF | 0.942 ± 0.039 | 0.969 ± 0.002 | 0.955 ± 0.021 | 0.969 ± 0.002 | 0.973 ± 0.012 | 0.965 ± 0.012 |
| | MoECF–AF | **0.943** ± **0.040** | 0.972 ± 0.004 | **0.957** ± **0.023** | **0.972** ± **0.004** | **0.981** ± **0.014** | **0.980** ± **0.014** |
| YELPCHI | GCNSVD | 0.686 ± 0.016 | 0.685 ± 0.016 | 0.685 ± 0.016 | 0.685 ± 0.016 | 0.824 ± 0.016 | 0.828 ± 0.016 |
| | NetMF | 0.695 ± 0.005 | 0.683 ± 0.013 | 0.678 ± 0.018 | 0.683 ± 0.013 | 0.696 ± 0.017 | 0.646 ± 0.017 |
| | GraRep | 0.721 ± 0.059 | 0.702 ± 0.034 | 0.697 ± 0.027 | 0.702 ± 0.034 | 0.731 ± 0.028 | 0.669 ± 0.028 |
| | FEATHER–N | 0.796 ± 0.035 | 0.796 ± 0.036 | 0.796 ± 0.036 | 0.796 ± 0.036 | 0.880 ± 0.035 | 0.866 ± 0.035 |
| | MoECF | 0.840 ± 0.004 | 0.837 ± 0.006 | 0.837 ± 0.006 | 0.837 ± 0.006 | 0.888 ± 0.006 | 0.877 ± 0.006 |
| | MoECF–AF | **0.848** ± **0.008** | **0.848** ± **0.008** | **0.848** ± **0.008** | **0.848** ± **0.008** | **0.913** ± **0.008** | **0.901** ± **0.008** |
| BITCOIN | GCNSVD | 0.840 ± 0.018 | 0.872 ± 0.029 | 0.855 ± 0.005 | 0.852 ± 0.001 | 0.924 ± 0.004 | 0.917 ± 0.004 |
| | NetMF | 0.915 ± 0.022 | 0.896 ± 0.011 | 0.905 ± 0.005 | 0.906 ± 0.006 | 0.945 ± 0.006 | 0.934 ± 0.006 |
| | GraRep | 0.913 ± 0.011 | 0.885 ± 0.005 | 0.898 ± 0.003 | 0.900 ± 0.004 | 0.943 ± 0.004 | 0.932 ± 0.004 |
| | FEATHER–N | 0.932 ± 0.001 | 0.910 ± 0.001 | 0.921 ± 0.001 | 0.922 ± 0.001 | 0.958 ± 0.001 | 0.944 ± 0.001 |
| | MoECF | 0.930 ± 0.019 | 0.942 ± 0.017 | 0.935 ± 0.001 | 0.935 ± 0.003 | 0.971 ± 0.003 | 0.961 ± 0.003 |
| | MoECF–AF | **0.954** ± **0.001** | **0.954** ± **0.001** | **0.954** ± **0.001** | **0.954** ± **0.001** | **0.981** ± **0.001** | **0.972** ± **0.001** |

graphs such as YelpChi, where heterophily and attribute polymorphism dominate, the attribute-only path is already discriminative, but fusion further improves ROC and PR behavior by regularizing expert routing under neighbor-label discordance and down-weighting spurious cross-view discrepancies; and on cryptocurrency graphs like Bitcoin, where long-range transactional dependencies make multi-hop factorization models competitive, the fused model remains dominant by preserving global structure through CF signatures while sharpening decision boundaries through adaptive attribute specialization. Taken together, these results indicate that our proposed methods generalize across node features with structural relation by leveraging distributional features, role-aware attribute encoding, and adaptive fusion for explicit conflict resolution, achieving calibration and discriminability in the fraud node classification performance most relevant for fraud detection pipelines.

Table 3: Ablation on variant and specialization. We report F1 and AUPRC for the four baseline fraud graphs.

| Variant | SC–TF | | BUPT–TF | | YelpChi | | Bitcoin | |
|---|---|---|---|---|---|---|---|---|
| | F1 | AUPRC | F1 | AUPRC | F1 | AUPRC | F1 | AUPRC |
| SVD only | 0.853 | 0.939 | 0.944 | 0.977 | 0.801 | 0.862 | 0.920 | 0.945 |
| SVD + MoE (MoECF) | 0.863 | 0.939 | 0.955 | 0.965 | 0.837 | 0.877 | 0.935 | 0.961 |
| SVD + MoE (mean only, $\lambda{=}0$) | 0.868 | 0.942 | 0.956 | 0.972 | 0.841 | 0.888 | 0.942 | 0.966 |
| SVD + MoE + Fuse (Uniform-MoE $\lambda{=}0.5$) | 0.869 | 0.942 | 0.956 | 0.977 | 0.843 | 0.892 | 0.946 | 0.968 |
| SVD + MoE + Fuse (MoECF–AF) | **0.873** | **0.944** | **0.957** | **0.980** | **0.848** | **0.901** | **0.954** | **0.972** |

An extended evaluation is provided in Appendix A.1, where we complement the aggregate metrics with ranking-oriented diagnostics (PR/ROC curves, Hit Rate, Precision@k) to assess calibration and retrieval quality under practically relevant evaluation settings.

**Ablation study** Table 3 isolates the mechanism for the ablation study: starting with SVD-only structural summaries, adding the MoE attribute path yields consistent cross-domain uplift, evidencing non-redundant features from role-conditioned attributes under polymorphism. Naïve fusion including global weighting or averaging offers limited gains beyond MoE, indicating that uniform mixing underfits cross-node variability and discards informative cross-modal discrepancy. In contrast, the adaptive fusion variant establishes the relative optimal in F1 and AUPRC on every dataset, demonstrating that instance-wise, discrepancy-aware mixing is essential for calibrated discrimina-

Table 4: Performance by explicit local-homophily bins on the test split. For each bin, we report F1 and AUPRC for structure-only (SVD-only), concatenation (MoECF), and fusion (MoECF–AF).

| Variant (homophily bin) | SC–TF | | BUPT–TF | | YelpChi | | Bitcoin | |
|---|---|---|---|---|---|---|---|---|
| | F1 | AUPRC | F1 | AUPRC | F1 | AUPRC | F1 | AUPRC |
| Bin: [0, 0.4] | | | | | | | | |
| SVD-only | 0.838 | 0.930 | 0.934 | 0.971 | 0.776 | 0.852 | 0.916 | 0.946 |
| MoECF | 0.848 | 0.934 | 0.946 | 0.973 | 0.814 | 0.872 | 0.934 | 0.960 |
| MoECF–AF (fused) | **0.858** | **0.938** | **0.948** | **0.975** | **0.828** | **0.888** | **0.942** | **0.966** |
| Bin: (0.4, 0.7] | | | | | | | | |
| SVD-only | 0.856 | 0.939 | 0.946 | 0.978 | 0.808 | 0.868 | 0.928 | 0.952 |
| MoECF | 0.866 | 0.942 | 0.958 | 0.980 | 0.844 | 0.892 | 0.944 | 0.965 |
| MoECF–AF (fused) | **0.876** | **0.947** | **0.961** | **0.983** | **0.854** | **0.908** | **0.958** | **0.974** |
| Bin: (0.7, 1.0] | | | | | | | | |
| SVD-only | 0.854 | 0.938 | 0.944 | 0.977 | 0.804 | 0.870 | 0.931 | 0.954 |
| MoECF | 0.864 | 0.940 | 0.956 | **0.982** | 0.846 | 0.888 | 0.946 | 0.968 |
| MoECF–AF (fused) | **0.870** | **0.943** | **0.956** | 0.981 | **0.846** | **0.900** | **0.953** | **0.971** |

tion under heterophily. Overall, the insight is that specialization (MoE) plus per-node fusion drives the observed performance and transfer across telecom, e-commerce, and cryptocurrency graphs.

**Evaluation by local homophily.** We compute a per-node local homophily score on the test split, defined as the fraction of a node's neighbours sharing its label, and evaluate models in homophily bins. This score is a proxy for the reliability of structural context: high homophily implies that the neighborhood supports the node's label, whereas low homophily implies that the neighborhood suggests the opposite label. Consequently, low-homophily bins capture cases where the structural feature is misleading for the node, while attribute features remain node-specific; i.e., structure and attributes are more likely to disagree. Binning by local homophily therefore tests whether the fusion mechanism can reconcile a supportive versus contradiction relative to the node's label.

We partition test nodes into homophily bins and measure F1/AUPRC per bin for structure-only, concatenation, and fusion variants. The fused model consistently achieves the largest gains in the lowest homophily pattern, i.e., where structural neighborhoods disagree with node labels and attribute features dominate; the margin narrows as homophily increases, indicating that the advantage stems from resolving cross-modal conflict rather than exploiting easy structure. MoECF generally exceeds structure-only in the heterophilous bins, yet the adaptive fusion model remains improved results, showing that the fusion improved handling of structure–attribute conflicts.

## 5 CONCLUSION

We present Mixture-of-Experts Characteristic Function (MoECF) node representation learning, a decouple–then–fuse method for fraud detection on heterogeneous graphs. The method isolates attribute and structural channels, equips them with modality-specific encoders, and reconciles their outputs through a per-node consensus–difference fusion with adaptive weighting. The approach addresses the brittleness of prior entangled approaches under structural heterophily and attribute polymorphism, exposing cross-modal disagreement as a discriminative signal. Empirical results across telecom, e-commerce, and cryptocurrency graphs demonstrate consistent gains in calibration and classification, highlighting the value of compositional representations for domain-agnostic fraud detection. This suggests that the proposed method provides a promising basis for building more generalisable fraud detectors capable of transferring across domains with reduced adaptation.

**Limitation** The proposed method could retain computational bottlenecks in conditions. In particular, dense graphs, excessive characteristic-function orders or evaluation points, and weak MoE sparsity can inflate both memory and runtime, making training infeasible at scale. These scenarios are analyzed in detail in Appendix A.3.3, which formalizes the conditions under which the pipeline exceeds device capacity and motivates mitigations such as low-rank truncation, sparse multiplies, and strict top-$k$ routing. We note that these constraints primarily arise in extreme settings, whereas in the benchmark graphs studied here, the method remains computational feasible and stable.

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

# A APPENDIX

## A.1 EXTENDED EVALUATION

We evaluate model performance using four complementary ranking diagnostics, applied consistently across all datasets: (i) Precision–Recall (PR) curves, (ii) Receiver Operating Characteristic (ROC) curves, (iii) Hit Rate, defined as local recall at varying top-percentile thresholds, and (iv) Precision at rank, measured at multiple cutoff levels within the low-to-mid percentile range. All evaluations are conducted under the standardized testing configuration described in Section 4, which enforces identical data splits, consistent thresholding procedures, and nonlinear scaling of the $y$-axes in PR and ROC plots to improve resolution in the high-performance clearance.

Across all four datasets (Figure 1), the fused model consistently demonstrates curve dominance in both PR and ROC: its PR trace sustains higher precision across nearly the entire recall range, while its ROC shows stronger true-positive yield in the small FPR pattern. This improved performance is reflected locally, as Hit Rate curves reveal a higher concentration of positives in the top-ranked deciles, and Precision@k remains uniformly higher across cutoffs. Notably, the MoE-only variant inherits some of these gains by specializing to attribute patterns, but without the structural correction from CF, its improvement plateaus earlier; FeatherNode, while robust among structural baselines, consistently underperforms the fused model in both global and head metrics.

Across datasets, a consistent pattern emerges: in telecom graphs (SC–TF, BUPT–TF), where neighborhood distributions are comparatively stationary (i.e., low churn and stable call semantics), FEATHER–N already attains strong PR/ROC; nevertheless, MoECF–AF delivers additive lift by injecting role-conditioned attribute specialization atop structure, with SC–TF exhibiting PR dominance, most visible ROC gains at small FPR, and sharper head concentration in Precision@k; in BUPT–TF, NetMF/GraRep inflate recall via higher-order proximity yet collapse in calibration (AUPRC), whereas MoECF–AF sustains both recall and precision, indicating effective reconciliation of structural regularities with skewed priors. In YelpChim, the concatenation path is already competitive (i.e., attribute-pattern separation is discriminative), but fusion improves ranking calibration: PR maintains higher precision at low recall and decays more gracefully toward full recall, ROC shifts upward across the FPR axis, and Hit Rate plus Precision@k confirm stronger head concentration. Finally, on Bitcoin, where long-range dependencies prevail, factorization methods (NetMF, GraRep) achieve high recall yet exhibit the recall–precision asymmetry; MoECF–AF preserves global structure via CF signatures while affecting the decision boundaries through adaptive

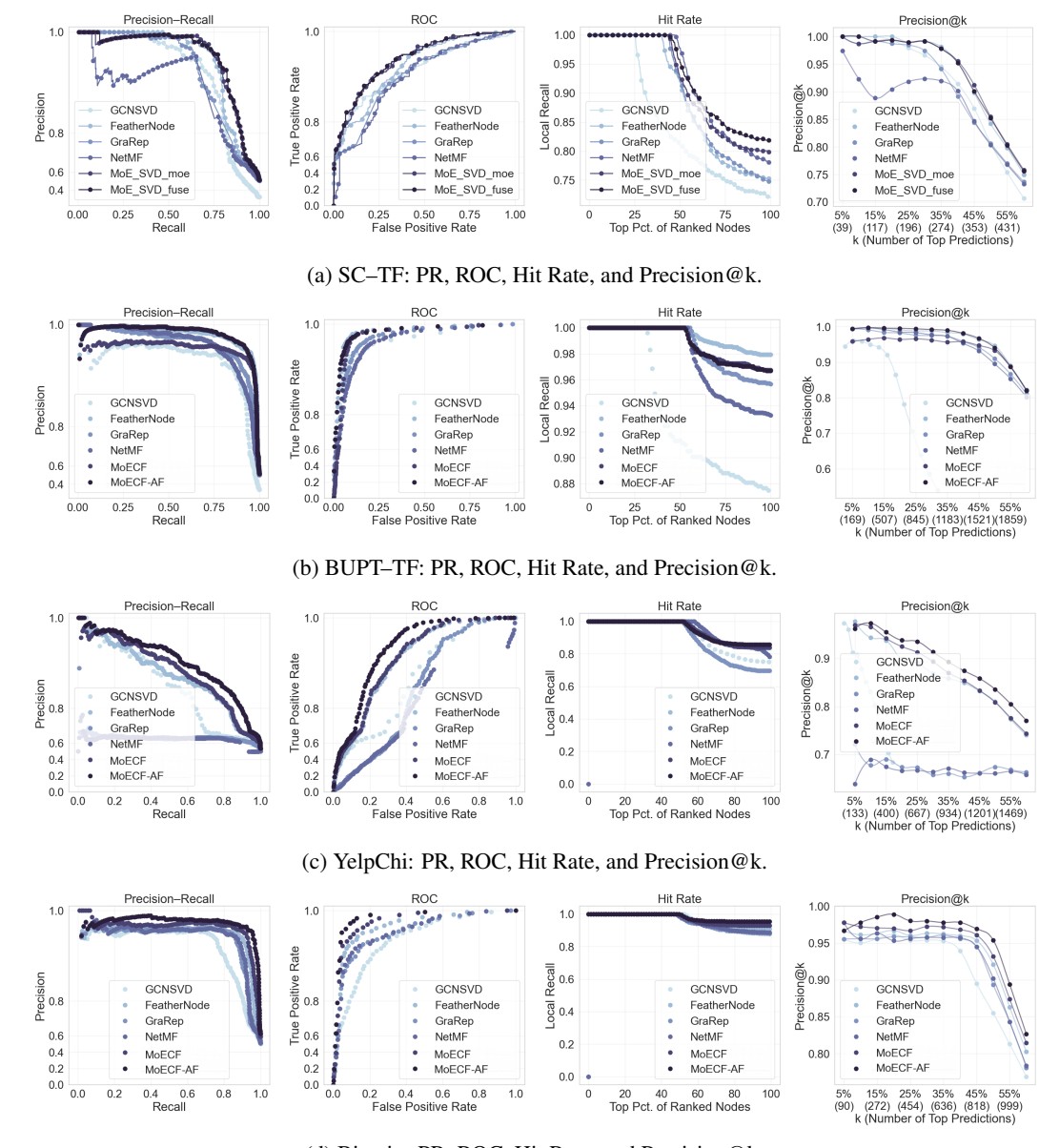

(a) SC–TF: PR, ROC, Hit Rate, and Precision@k.

(b) BUPT–TF: PR, ROC, Hit Rate, and Precision@k.

(c) YelpChi: PR, ROC, Hit Rate, and Precision@k.

(d) Bitcoin: PR, ROC, Hit Rate, and Precision@k.

Figure 1: Evaluation curves on four datasets: telecom (SC–TF, BUPT–TF), e-commerce (YelpChi), and cryptocurrency (Bitcoin).

attributes, yielding consistent upward shifts in PR/ROC, especially in the small FPR operating region and in Precision@k, whereas GCNSVD, though resilient under local homophily, deteriorates under heterophily, evidencing the brittleness of isotropic aggregation in such patterns.

Matrix factorization baselines consistently exhibit a recall–precision asymmetry: aggressive coverage inflates recall but depresses precision, leading to diminished AUPRC and weaker head concentration, most visible on BUPT–TF and Bitcoin. GCNSVD retains competitiveness on telecom graphs where homophily is stronger but degrades on YelpChi and Bitcoin, underscoring the limitations of smoothing-based propagation under heterophily and structural discordance. In contrast, the fusion model sustains both stability and calibration across domains, with ROC/PR curves that degrade more gracefully and Precision@k consistently higher across all rank thresholds.

## A.2 MoE Hyper-Parameter Sensitivity

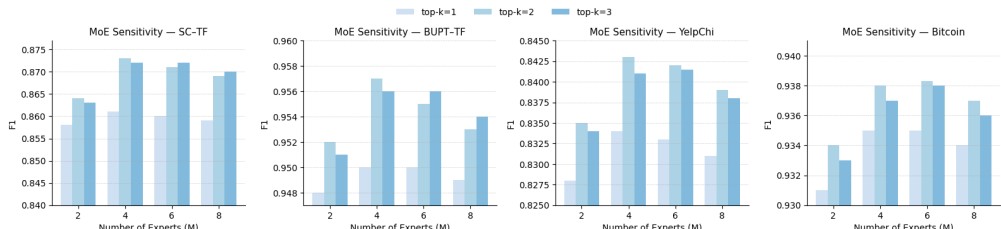

Figure 2: MoE hyper-parameter sensitivity on SC–TF, BUPT–TF, YelpChi, and Bitcoin. Same-color grouped bars emphasize discrete choices.

When the number of experts is limited, the model cannot adequately capture diverse attribute conditions, which leads to underfitting . Increasing the number of experts alleviates this issue, but beyond a moderate size the gains diminish and redundancy begins to dominate. top-k determines how many experts contribute to each input, where too few induce brittle reliance on a single projection and too many blur specialization through excessive mixing. Our experiments show that an intermediate setting (M=4, top-k=2) of both the expert pool and routing achieves the most stable and discriminative representations as Figure 2 shows. We therefore adopt this setting for all subsequent experiments.

## A.3 Theoretical Analysis

We inherit the notation of Section 3. Let $G = (V, E)$ be a graph with $N = |V|$ nodes, adjacency $\mathbf{A}$, degree matrix $\mathbf{D}$, and row-stochastic walk matrix $\tilde{\mathbf{A}} = \mathbf{D}^{-1}\mathbf{A}$ (i.e., one-step random-walk transition). Node $v_i$ has raw attributes $\mathbf{x}_i \in \mathbb{R}^d$; the MoE encoder (Section 3.2) outputs $\mathbf{z}_i \in \mathbb{R}^{d'}$. For fusion (Section 3.4), let $\mathbf{e}_i \in \mathbb{R}^r$ denote the $i$-th row of the structural code $\mathbf{E} \in \mathbb{R}^{N \times r}$ (Section 3.3). We form PCA-aligned, whitened views $\tilde{\mathbf{e}}_i = \mathrm{whiten}(\mathbf{e}_i \mathbf{P}_E) \in \mathbb{R}^{d^\star}$ and $\tilde{\mathbf{z}}_i = \mathrm{whiten}(\mathbf{z}_i \mathbf{P}_Z) \in \mathbb{R}^{d^\star}$, with $d^\star = \min(r, d')$. Define per-node consensus $\mathbf{h}_i$ and discrepancy $\mathbf{d}_i$:

$$\mathbf{h}_i = \tfrac{1}{2}(\tilde{\mathbf{e}}_i + \tilde{\mathbf{z}}_i), \qquad \mathbf{d}_i = \tilde{\mathbf{z}}_i - \tilde{\mathbf{e}}_i,$$

and the fused attribute code $\mathbf{f}_i = \left[\, \mathbf{h}_i^\top \mid (\lambda_i \mathbf{d}_i)^\top \,\right]^\top \in \mathbb{R}^{2d^\star}$, where $\lambda_i \in [0, 1]$ is the per-node weight produced by the variational mixer (Section 3.5). For the structural channel (Section 3.3), let $\Theta = \{\theta_\ell\}_{\ell=1}^{L} \subset \mathbb{R}^{d^\star}$ be frequency vectors and define the CF lift:

$$\phi_\Theta(\tilde{\mathbf{x}}) = \left[\, \cos(\tilde{\mathbf{x}}^\top \theta_1), \sin(\tilde{\mathbf{x}}^\top \theta_1), \ldots, \cos(\tilde{\mathbf{x}}^\top \theta_L), \sin(\tilde{\mathbf{x}}^\top \theta_L) \,\right]^\top \in \mathbb{R}^{2L}.$$

Stacking $\mathbf{\Phi} \in \mathbb{R}^{N \times 2d^\star L}$ by applying $\phi_\Theta$ to each node and diffusing up to order $O$ yields $\mathbf{H} = \left[\tilde{\mathbf{A}}\mathbf{\Phi} \parallel \tilde{\mathbf{A}}^2\mathbf{\Phi} \parallel \cdots \parallel \tilde{\mathbf{A}}^O\mathbf{\Phi}\right] \in \mathbb{R}^{N \times (2d^\star LO)}$. Randomized truncated SVD provides $\mathbf{H} \approx \mathbf{U}_r \mathbf{S}_r \mathbf{V}_r^\top$ and we retain $\mathbf{E} = \mathbf{U}_r \mathbf{S}_r \in \mathbb{R}^{N \times r}$ as the structural code. In words, per-node CF features are constructed from PCA-aligned, whitened attributes via cosine/sine evaluations at $L$ frequencies and then propagated through 1:$O$ powers of the random-walk matrix to form a multi-hop, distributional neighborhood stack $\mathbf{H}$; we then implement a randomized truncated SVD accessing $\mathbf{H}$ only to obtain a compact structural code $\mathbf{E} \in \mathbb{R}^{N \times r}$ that preserves the dominant, permutation-invariant features while remaining scalable on graphs.

### A.3.1 Characteristic-Function Neighborhood Encoding

The CF neighborhood encoder admits (i) distributional separability, (ii) finite-$L$ concentration, and (iii) stable, energy-preserving diffusion and compression; together these yield permutation-invariant, multi-hop distributional embeddings that discriminate node contexts under heterophily while remaining statistically stable and computationally scalable.

**Separability.** Let $\mathbb{P}, \mathbb{Q}$ be two distinct neighborhood attribute distributions (short random-walk contexts around different node types) with characteristic functions $\Psi, \Phi$. If $\mathbb{P} \neq \mathbb{Q}$, then $\Psi \not\equiv \Phi$, so CF features at finitely many random frequencies discriminate neighborhood types with probability one. This induces embeddings whose geometry reflects differences between neighborhood distributions, enabling practical separation of benign vs. fraudulent contexts even under heterophily.

**Concentration.** For whitened attributes $\tilde{\mathbf{x}}$ with $\|\tilde{\mathbf{x}}\|_2 \leq B$ (uniform upper bound on the Euclidean norm of the whitened attribute vectors), the empirical CF vector using $L$ i.i.d. frequencies concentrates around its expectation at rate $O(1/\sqrt{L})$ (uniformly with high probability), i.e., a modest $L$ yields stable CF summaries. In practice, tens of frequencies suffice to obtain low-variance codes across resampling windows or stochastic data slices, supporting reproducible downstream training. Concentration is required because the empirical CF features are constructed from a finite sample of frequencies, and without concentration guarantees, the resulting embeddings could fluctuate substantially across different samples, undermining their statistical reliability and comparability.

**Stability and compression.** The diffusion stack $\mathbf{H} = [\tilde{\mathbf{A}}\boldsymbol{\Phi}\|\cdots\|\tilde{\mathbf{A}}^O\boldsymbol{\Phi}]$ is Lipschitz in $\tilde{\mathbf{A}}$ (i.e., small perturbations of the adjacency lead to bounded, proportionally small changes in $\mathbf{H}$). The rank-$r$ truncation of $\mathbf{H}$ minimizes Frobenius error and preserves a quantified fraction of signal (feature) energy, yielding a low-variance structural embedding $\mathbf{E} = \mathbf{U}_r\mathbf{S}_r$. The embedding is robust (i.e., small perturbations in the graph topology cause only limited, stable changes in the embedding) to minor topology noise and compresses to $r$ dimensions with controlled information loss, enabling efficient training on large, sparse fraud graphs.

### A.3.2 Mean–Difference Fusion with Per-Node Gating

Consensus as a Bayes estimator after whitening: Model the aligned views as $\tilde{\mathbf{e}}_i = \mathbf{s}_i + \boldsymbol{\epsilon}_x$ and $\tilde{\mathbf{z}}_i = \mathbf{s}_i + \boldsymbol{\epsilon}_z$ with independent, zero-mean, isotropic noise of comparable variance (the whitening step enforces scale comparability). Then the minimum-variance linear unbiased estimator of the shared signal $\mathbf{s}_i$ is the average: $\widehat{\mathbf{s}}_i = \frac{1}{2}(\tilde{\mathbf{e}}_i + \tilde{\mathbf{z}}_i) = \mathbf{h}_i$. In practice, $\mathbf{h}_i$ acts as a calibrated consensus code that preserves agreements between structure- and attribute-derived views while suppressing view-specific noise.

Information from disagreement and gated composition: Let $\mathbf{d}_i = \tilde{\mathbf{z}}_i - \tilde{\mathbf{e}}_i$ capture cross-view disagreement. For any smooth discriminative model $p_\theta(y \mid \cdot)$ built on the fused code, the Fisher information in $[\mathbf{h}_i^\top \mid (\lambda\mathbf{d}_i)^\top]^\top$ is no less than that in $\mathbf{h}_i$ alone; equality holds only if $\mathbf{d}_i$ is independent of the label or $\lambda = 0$. i.e., including a (possibly small) scaled discrepancy cannot harm and is beneficial whenever disagreement correlates with the label (e.g., attributes flag fraud while structure suggests benign neighbours). The discrepancy channel exposes cross-view conflicts that often carry a fraud signal and, when retained with a nonzero $\lambda$, sharpens decision boundaries under heterophily.

**Per-node trust weight.** Let $\lambda_i = \sigma(\mathbf{w}^\top\hat{\mathbf{d}}_i)$ be produced by the mixer; since $\sigma'(t) \leq 1/4$ (As $\sigma'(t) = \sigma(t)(1 - \sigma(t))$ attains its maximum at $t = 0$ with value $1/4$, the derivative is bounded as $\sigma'(t) \leq 1/4$ for all $t \in \mathbb{R}$) :

$$\left|\lambda_i(\hat{\mathbf{d}}_i) - \lambda_i(\hat{\mathbf{d}}_i')\right| \leq \tfrac{1}{4}\|\mathbf{w}\|_2\,\|\hat{\mathbf{d}}_i - \hat{\mathbf{d}}_i'\|_2,$$

so the gate is Lipschitz and numerically stable. i.e., the mixer adapts the retained amount of disagreement per node and avoids brittle flipping due to small estimation noise. In practice, $\lambda_i$ is for fusion to each node's context, upweighting informative conflicts and downweighting noisy ones, yielding calibrated scores across heterogeneous graphs.

### A.3.3 Computational Complexity of MoECF–AF

$N = |V|$ nodes, $E = |E|$ edges, raw attribute dim. $d$, MoE output dim. $d'$, aligned dim. $d^\star = \min(r, d')$, CF frequencies $L$, diffusion order $O$, SVD rank $r$, Krylov/Lanczos iterations $\eta$, number of experts $M$, active experts per node $k \ll M$, mixer size $p \ll d^\star$. Structural codes $\mathbf{E} \in \mathbb{R}^{N \times r}$ and MoE outputs $\mathbf{Z} \in \mathbb{R}^{N \times d'}$.

Attribute path (MoE): Routing logits over $M$ experts cost $O(N M d)$; evaluating the top-$k$ two-layer MLP experts of width $d'$ costs $O(N k d d')$.

$$\text{MoE time} = O(N M d) + O(N k d d') \quad (\text{with } k \ll M).$$

Structure path (CF $\rightarrow$ diffusion $\rightarrow$ SVD): CF lift over $d$ dims at $L$ frequencies costs $O(N d L)$. Randomized truncated SVD of rank $r$ with $\eta$ iterations accesses the diffusion stack via sparse multiplies only:

$$\text{CF lift time} = O(N d L), \quad \text{SVD time} = O\big((N + E)\,r\,\eta\big), \quad \text{SVD memory} = O(Nr).$$

Fusion and head: Per node, forming $\mathbf{h}_i, \mathbf{d}_i$ costs $O(d^\star)$; the mixer/gate costs $O(p)$; concatenation with $\mathbf{E}_i$ is $O(r + d^\star)$.

Total time per epoch:

$$O(N\,M\,d) + O(N\,k\,d\,d') + O(N\,d\,L) + O\big((N + E)\,r\,\eta\big) + O\big(N(r + d^\star)\big).$$

Dominant memory:

$$O(Nr) + O(Nd') \quad \text{(for } \mathbf{E} \text{ and } \mathbf{Z}) \ + \ \text{mini-batch activations.}$$

In practice, $O\big((N + E)\,r\,\eta\big)$ dominates on large sparse graphs (hence moderate $r, \eta$), while $k \ll M$ keeps MoE cost tractable.

**Limitation: when computation becomes infeasible.** Dense graphs ($E = \Theta(N^2)$); choosing $r$ near the stack width ($r \approx 2d^\star LO$, negating compression); weak MoE sparsity ($k \approx M$); or very large $L, O$ (inflating $d^\star L$ and SVD multiplies) can push time/memory beyond device limits (e.g., $Nr$ or $Nd'$ too large), motivating low-rank SVD, sparse multiplies, and top-$k$ routing.

