# OpenReview forum: "Mixture of Experts Characteristic Function Embeddings for Heterogeneous Fraud Graphs"
_ICLR.cc/2026/Conference — ICLR 2026 Conference Withdrawn Submission_

### Official Review · Reviewer_CKZw · 2025-10-28

**Soundness:** 3
**Presentation:** 2
**Contribution:** 2
**Rating:** 4
**Confidence:** 3

**Summary:**

This manuscript proposes a unified framework called MoECF–AF (Mixture-of-Experts Characteristic Function Adaptive Fusion) for fraud detection on heterogeneous graphs. The study targets the core challenge of entangled inductive bias between structural and attribute information, an issue that undermines model generalization under structural heterophily. To address this, the authors introduce a “decouple-then-fuse” representation paradigm that separately encodes structure and attributes before adaptively combining them.

**Strengths:**

- The strategy of decoupling and subsequently re-coupling the learning of attributes and structures is conceptually sound and theoretically capable of addressing the entangled inductive bias problem in heterogeneous graphs.

- The model achieves impressive experimental performance compared to the baselines.

- The experimental evaluation is thorough, including parameter sensitivity analyses and multiple ablation studies.

- The introduction of MoE top-k routing and randomized SVD enhances the model’s scalability and strengthens its practical applicability.

**Weaknesses:**

- The motivation for decoupling and re-coupling is not entirely clear. If existing methods perform well in both aspects, why is such a decouple-then-fuse process necessary?

- The notion of inductive bias entanglement remains largely conceptual and is not empirically demonstrated. The manuscript merely infers its resolution from the superior performance of MoECF-AF across datasets.

- Although the paper proposes a task-specific and meaningful fusion process, the decoupling process itself appears vague, as the model simply processes attributes and graph structures separately. Hence, the phrase “decouple-then-fuse” may be somewhat overstated.

- The manuscript claims superiority in cross-domain generalization, yet the experiments seem to train and test separately on each dataset, rather than using a single model across domains. This makes the “cross-domain” claim less convincing and similar to many existing fraud detection studies that simply perform well on multiple datasets.

- The Introduction section fails to clearly convey the study’s motivation and even omits the model’s name, which weakens the presentation.

- The baseline literature is relatively outdated and omits recent state-of-the-art fraud detection models, which may lead to overestimating the contribution of this work.

- It is recommended to add a model architecture figure to illustrate the overall design more clearly.

- The content organization could be improved; some important experiments should be moved from the appendix to the main text for better visibility and narrative coherence.

- The use of dashes in the manuscript appears inconsistent, please standardize their formatting.

**Questions:**

Please refer to the weaknesses.

---

### Official Review · Reviewer_iMsJ · 2025-10-29

**Soundness:** 1
**Presentation:** 1
**Contribution:** 1
**Rating:** 2
**Confidence:** 5

**Summary:**

This paper proposes a decouple-then-fuse representation learning framework for fraud detection on heterogeneous graphs, addressing challenges posed by relational multiplexity, attribute polymorphism, and structural heterophily. The method explicitly separates structural and attribute processing: the structural channel encodes distributional neighborhood context through characteristic-function signatures compressed via randomized spectral factorization, while the attribute channel employs MOE projections that provide input-adaptive specialization to role-conditioned patterns. Evaluating on four real-world fraud graphs demonstrates consistent improvements over baselines.

**Strengths:**

1. The benchmarks (Telecom, e-commerce, and cryptocurrency) provide meaningful diversity.
2. The "decouple-then-fuse" paradigm is intuitive.

**Weaknesses:**

1. In Eq. (4), the motivation and justification for the Fourier-like feature transformation are unclear. The connection to actual characteristic functions or Fourier analysis is never rigorously established, and no intuition is provided for why such a transformation would specifically benefit fraud detection. Most importantly, the paper lacks an ablation comparing this transformation to a simpler baseline that diffuses raw attributes directly before the SVD compression.
2. How $\theta_{\max}$ and $L$ are chosen is unclear.
3. Since cosine and sine are applied directly to raw $X$, the features are sensitive to attribute scale and units; no standardization/whitening is specified before Eq. (4), so the effective frequencies are arbitrary.
4. The paper's training protocol is fundamentally flawed. The MoE encoder is trained with a task head and then frozen; only afterwards is fusion applied. This decoupled design can be sub-optimal because the attribute encoder never sees gradients from the fusion/mixer, limiting synergy between views.
5. The top-k expert selection (Eq.(2)-(3)) introduces non-differentiability through hard routing (arg top-k), which zeros gradients to non-selected experts and risks expert collapse. The router computes softmax weights but then hard-selects k experts without the load-balancing or entropy regularization terms standard in MoE designs. This can lead to severe expert under-utilization.
6. The experimental setup is outdated and incomplete. The authors only compare against classical matrix-factorization baselines, which are no longer representative of current graph learning methods. Numerous heterophily-aware GNNs, such as H2GCN, GPR-GNN, and FAGCN, have been specifically proposed to address structural heterophily and should be included for a fair evaluation. Moreover, it is a serious omission that GNN-based fraud detection approaches like CARE-GNN, PC-GNN, and BWGNN are entirely excluded.
7. The paper does not follow the correct citation style required by the target venue. It consistently uses `\cite{}` instead of the recommended `\citep{}` command for parenthetical references. Besides, equations needs to be indexed.
8. The paper does not provide any source code or implementation details for the proposed model, which severely limits reproducibility. For top-tier conferences such as ICLR, public release of code is strongly encouraged to ensure transparency.

**Questions:**

See weaknesses.

---

### Official Review · Reviewer_J7gW · 2025-10-31

**Soundness:** 3
**Presentation:** 2
**Contribution:** 2
**Rating:** 4
**Confidence:** 3

**Summary:**

This paper proposes a novel decouple-then-fuse representation learning framework for fraud detection on heterophily graphs, addressing challenges like relational multiplexity, attribute polymorphism, and structural heterophily. It separates structural and attribute processing: structural embeddings use characteristic function signatures compressed via randomized spectral factorization, while attributes are handled by input-adaptive Mixture-of-Experts (MoE) projections. This paper uniquely combines MoE with characteristic functions and adaptive fusion, enhancing cross-domain generalization amid heterophily. The model outperforms baselines in fraud node detection.

**Strengths:**

S1. The manuscript employs clear, structured academic prose with logical flow from problem motivation to method details and experiments. Sections are well-organized, with precise definitions and motivations.

S2. The fusion mechanism innovatively integrates Bayesian principles with variational mixing, providing theoretical justification via ELBO-inspired objectives.

**Weaknesses:**

W1: While core components are described, code availability is not mentioned, and specifics on training environments (e.g., hardware, random seeds) are absent.

W2: Some sections, such as the related work, list references densely without deep critical analysis of recent 2025 advances, leading to superficial integration.

W3: The adaptive λ discretization is noted but lacks justification for grid size or sensitivity analysis.

W4: The experimental evaluation employs outdated baselines (with the most recent published in 2020), failing to adequately represent recent advancements (e.g., [1-4]) in heterophily-tolerant GNNs for fraud detection. This constrains the evaluation of the proposed model’s innovation.

W5: The paper conflates "heterogeneous" with "heterophily" by including structural heterophily as an aspect of graph heterogeneity, which may confuse readers familiar with definitions where heterogeneous graphs refer to multi-type nodes/edges, and heterophily denotes label/feature dissimilarity in connected nodes. By subsuming heterophily under heterogeneity, the model’s robustness claims may overlook distinct challenges, such as heterophily in homogeneous subgraphs within heterogeneous graphs. This could limit theoretical depth compared to works like Hetero2Net [5], which separately address both.

Refs.:

[1] Xu, Fan, et al. "Revisiting graph-based fraud detection in sight of heterophily and spectrum." Proceedings of the AAAI conference on artificial intelligence. 2024.

[2] Gao, Yuan, et al. "Addressing heterophily in graph anomaly detection: A perspective of graph spectrum." Proceedings of the ACM web conference 2023. 2023.

[3] Fu, Chao, et al. "Nowhere to H 2 IDE: Fraud Detection From Multi-Relation Graphs via Disentangled Homophily and Heterophily Identification." IEEE Transactions on Knowledge and Data Engineering (2024).

[4] Shi, Fengzhao, et al. "H2-fdetector: A gnn-based fraud detector with homophilic and heterophilic connections." Proceedings of the ACM web conference 2022. 2022.

[5] Li, Jintang, et al. "Hetero2 Net: Heterophily-aware Representation Learning on Heterogenerous Graphs." arXiv preprint arXiv:2310.11664 (2023).

**Questions:**

W1 and W5: Suggest providing source code for community replication and checking the use of technical terminology.

W2 and W4: Discussion and experimentation (if not necessary, please explain) on the recent advancements.

W3: Discussion about the impact of the adaptive λ.

---

### Official Review · Reviewer_L6Ks · 2025-11-09

**Soundness:** 2
**Presentation:** 3
**Contribution:** 3
**Rating:** 4
**Confidence:** 3

**Summary:**

The authors propose a "decouple-then-fuse" representation learning framework for fraud detection on heterogeneous graphs, which simultaneously exhibit conflicting structural patterns (heterophily) and diverse node features (attribute polymorphism). The model first encodes graph structure using characteristic-function signatures, which are robust to heterophily, and separately encodes node attributes using a Mixture-of-Experts (MoE) model to specialize in different node roles . These two distinct views are then reconciled through a Bayesian mean-difference fusion layer, which adaptively learns a per-node weight to combine the "consensus" and "discrepancy" between the structural and attribute signals . This method demonstrates improved fraud detection performance across telecom, e-commerce, and cryptocurrency datasets by effectively resolving conflicts between the two modalities.

**Strengths:**

There are a few things I like about the paper:
1. The paper addresses a challenging and practical problem in heterogeneous graph fraud detection.
2. The "decouple-then-fuse" strategy is an interesting and well-motivated design choice. It hopes to avoid the "premature entanglement" of structural and attribute biases, where the authors stated that it’s the limitation of existing monolithic models..
3. The Bayesian fusion mechanism is also interesting. It models consensus and discrepancy, and then learns a per-node weight to combine them.
4. The authors demonstrated improvement over the baselines on multiple baselines.

**Weaknesses:**

1. [Baselines]. The paper did not compare against more recent graph neural networks models specifically designed for handling heterogeneous graphs, such as R-GCN, HGT, and their more recent variations.
2. [Complexity] The proposed pipeline is significantly complex, involving multiple distinct stages.
3. [Hyperparameters] The full model has a large number of hyperparameters. While some sensitivity analyses are explored, the tuning process for the others is not discussed.

**Questions:**

Please address the weaknesses mentioned above.

---

### Note · Authors · 2025-11-15

I have read and agree with the venue's withdrawal policy on behalf of myself and my co-authors.